# Tobacco Plastid Transformation as Production Platform of Lytic Polysaccharide MonoOxygenase Auxiliary Enzymes

**DOI:** 10.3390/ijms24010309

**Published:** 2022-12-24

**Authors:** Rachele Tamburino, Daniela Castiglia, Loredana Marcolongo, Lorenza Sannino, Elena Ionata, Nunzia Scotti

**Affiliations:** 1CNR-IBBR, Institute of Biosciences and BioResources, 80055 Portici, Italy; 2CNR-ICB, Institute of Biomolecular Chemistry, 80078 Pozzuoli, Italy; 3CNR-IRET, Research Institute on Terrestrial Ecosystems, 80131 Naples, Italy

**Keywords:** chloroplast transformation, lignocellulosic biomass, biorefinery, family AA9, auxiliary activities, carbohydrate-modifying enzymes

## Abstract

Plant biomass is the most abundant renewable resource in nature. In a circular economy perspective, the implementation of its bioconversion into fermentable sugars is of great relevance. Lytic Polysaccharide MonoOxygenases (LPMOs) are accessory enzymes able to break recalcitrant polysaccharides, boosting biomass conversion and subsequently reducing costs. Among them, auxiliary activity of family 9 (AA9) acts on cellulose in synergism with traditional cellulolytic enzymes. Here, we report for the first time, the production of the AA9 LPMOs from the mesophilic *Trichoderma reesei* (TrAA9B) and the thermophilic *Thermoascus aurantiacus* (TaAA9B) microorganisms in tobacco by plastid transformation with the aim to test this technology as cheap and sustainable manufacture platform. In order to optimize recombinant protein accumulation, two different N-terminal regulatory sequences were used: 5′ untranslated region (5′-UTR) from *T7g10* gene (DC41 and DC51 plants), and 5′ translation control region (5′-TCR), containing the 5′-UTR and the first 14 amino acids (Downstream Box, DB) of the plastid *atpB* gene (DC40 and DC50 plants). Protein yields ranged between 0.5 and 5% of total soluble proteins (TSP). The phenotype was unaltered in all transplastomic plants, except for the DC50 line accumulating AA9 LPMO at the highest level, that showed retarded growth and a mild pale green phenotype. Oxidase activity was spectrophotometrically assayed and resulted higher for the recombinant proteins without the N-terminal fusion (DC41 and DC51), with a 3.9- and 3.4-fold increase compared to the fused proteins.

## 1. Introduction

In recent years, the circular economy model based on biorefinery platforms that endeavor to reduce waste and increase productivity has had a huge boost. In this new model, plant biomasses are highly significant as a source of energy and high value products. It has been estimated that agricultural activities produce several billion tons of lignocellulosic residues. Although lignocellulosic plant biomasses consist mainly of three types of polymers (i.e., lignin, cellulose and hemicellulose), the proportion of each polymer varies between residues of diverse origin. Despite their wide application, it is well known that plant biomass breakdown requires a complex enzymatic machinery composed of different classes of cell wall degrading enzymes [1,2]. Recently, great attention has been focused on a novel class of auxiliary enzymes known as Lytic Polysaccharide MonoOxygenases (LPMOs) able to break recalcitrant polysaccharide [2,3,4]. Actually, LPMOs, which are mono copper-dependent enzymes, are classified into eight families AA9, AA10, AA11, AA13, AA14, AA15, AA16 and AA17 [2,5,6]. Among them, the AA9, formerly known as GH61, represents a fungal family that acts on cellulose in synergism with traditional cellulolytic enzymes. Rani Singhania, et al. [4] demonstrated that LPMOs, when added to a cellulolytic cocktail, produced a significant increase in the rate and efficiency of cellulose deconstruction. Indeed, these enzymes, through their oxidative activity, introduce chain breaks into the crystalline regions of recalcitrant polysaccharides, such as cellulose and chitin, leading to the formation of oxidized products. This mechanism is the basis of the boosting action of LPMOs that makes the polysaccharides more hydrolysable by glycosidases. Thus, by increasing the accessibility of the substrate, LPMOs enhance the overall efficiency of the degradation, improving the enzymatic conversion of biomass [7]. Notwithstanding the great interest toward this class of enzymes, their functional characterization remains a major challenge due to the absence of a specific assay to test the enzymatic activity and the induction of side reactions, including auto-catalytic inactivation of the enzyme [8]. To date, most of the characterized LPMOs were recombinantly produced only in microbial hosts such as *Escherichia coli*, *Pichia pastoris* and fungal strains [8,9].

Over the past decades, it has been demonstrated that plants represent a promising system to produce recombinant proteins due to low production costs and availability of several strategies to optimize yield and overcome limitations [10]. Among them, the chloroplast transformation platform shows some attractive advantages (e.g., site-specific gene integration, absence of epigenetic gene silencing and position effect, compartmentalization of recombinant protein, possibility to express multiple genes arranged as operon, transgene containment due to maternal inheritance of plastid in most crops) and allows extraordinarily high production yield for various recombinant proteins [11,12,13], including cellulolytic enzymes (up to ≥75% of Total Soluble Proteins, TSP) [14,15,16].

The aim of this study was to explore, for first time, the feasibility of plants and, in particular, chloroplast transformation technology as a heterologous platform to produce in tobacco two synthetic Lytic Polysaccharide MonoOxygenases, belonging to AA9 family and derived from the mesophilic *Trichoderma reseei* (termed herein TrAA9B) and the thermophilic *Thermoascus auranticus* (termed herein TaAA9B) fungi. Indeed, the availability of a green, low-cost and sustainable manufacture platform will allow to overcome one of the major limitations in application of enzymes in biomass conversion reducing production and processing costs. Different expression levels and protein stability were observed for the two LPMO enzymes. Further, the enzymatic assay results demonstrated oxidative activities for the plastid-based LPMO enzymes.

## 2. Results

### 2.1. Vector Construction and Production of LPMOs Transplastomic Plants

To transform tobacco plastid genomes, four vectors containing the mesophilic (*TrAA9B*) and thermophilic (*TaAA9B*) LPMOs were constructed. For each transgene, different 5′ regulatory sequences were used (Figure 1). In particular, all constructs contain the same constitutive ribosomal RNA operon promoter (P*rrn*) and the 3′ untranslated region (3′-UTR) of the plastid *rbcL* gene (T*rbcL*).

Further, each coding sequence was fused either with the 5′ untranslated region (5′-UTR) from the gene *10* of *E. coli* phage T7 (*T7g10*) in pDC41 and pDC51 vectors or with a translational control region (5′-TCR) containing the 5′-UTR from *atpB* and 42 N-terminal nucleotides (downstream box, DB) of *atpB* coding sequence in pDC40 and pDC50 vectors. A Flag tag was added at C-terminus of all coding regions for protein detection. The expression cassettes target exogenous DNA to the *trnV-rps12/7* region of the tobacco plastid genome and contain *aminoglycoside 3′ adenylyltransferase* (*aadA*) gene as a selectable marker (Figure 1). The biolistic method was used to transform tobacco plants producing several primary spectinomycin-resistant shoots (12–15) for each vector that were further selected on medium containing either spectinomycin or spectinomycin and streptomycin to eliminate spontaneous spectinomycin-resistant mutants and regenerate secondary spectinomycin-resistant shoots (8–13 for each vector). Homoplasmy and the correct integration of foreign DNA were verified by Southern blot analysis with the *trnV-rps12/7* targeting sequence as the probe (Figure 2a). Fragments of 5.8 kb and 3.3 kb were identified for all LPMO transplastomic and wild-type plants, respectively (Figure 2b).

### 2.2. Phenotypes of AA9 LPMO Transplastomic Plants

Transplastomic plants grown in soil under greenhouse conditions generally showed a phenotype similar to that observed in wild type (Nt) and transformed with empty vector (PRV) control plants (Figure 3). Only DC50 transplastomic plants exhibited a retarded growth and a mild pale-green phenotype compared to the controls. Such difference was clearly perceived at both of the early stages of development (18 days after transplanting, DAT) and 46 DAT, when the other plants displayed flowers. Despite its retarded growth, DC50 transplastomic plants were able to reach maturity (90 DAT), flower and produce viable seed by selfing similarly to the other transplastomic and control plants.

### 2.3. Expression Analyses of AA9 LPMO Transplastomic Plants

Expression analyses of TaAA9B and TrAA9B transgenes were carried out on T0, T1 and T2 transplastomic plants at both transcription and protein levels. Total RNA was extracted from leaves of plants before flowering stage. Semi-quantitative (27 cycles) RT-PCR amplification of either TaAA9B or TrAA9B genes showed the expected 0.5 kb PCR product (Figure 4). For both transgenes, a higher transcript level was observed in transplastomic plants produced with vectors containing the 5′-TCR (pDC40 and pDC50), whereas the results obtained with the housekeeping gene 18S clearly demonstrate loading controls.

Western blot analyses with N-terminal Flag fusion protein of *E. coli* bacterial alkaline phosphatase (FlagBAP) as reference revealed different accumulation levels for the recombinant proteins (Figure 5). Particularly, the mesophilic TrAA9B expressed as fusion protein with the N-terminal 14 amino acids of the plastid protein AtpB (DC50) showed, accordingly with RT-PCR results, the highest yield (5% TSP), whereas the same protein without the N-terminal fusion (DC51) resulted in lower accumulation level (0.5% TSP). On the contrary, the thermophilic form reached highest level of accumulation when expressed under control of T7g10 5′-UTR (DC41, 1% TSP), while in DC40 plants, where TaAA9B was expressed as fused protein at its N-terminal, a slightly lower protein yield (0.7% TSP) was obtained.

In order to evaluate the in planta stability of the recombinant proteins, Western blot analyses were also performed using leaves at different developmental stages (Figure 6). DC50 transplastomic plants showed the highest stability since the accumulation profile was unaltered along the leaf gradient; on the contrary, DC40 largely accumulated at early stage of leaf development being present only in the third and the fifth leaves, whilst in DC41 and DC51 transplastomic plants, the recombinant proteins were detected in mature leaves (fifth and seventh).

### 2.4. LPMO Activity

Lytic polysaccharide MonoOxygenases are able to convert 2,6-dimethoxyphenol (2,6-DMP) into coerulignone [17]. In particular, the Cu II containing active site of LPMO catalyzes the oxidation of 2,6-DMP, which is converted into 2,6-DMP radicals, with subsequent reduction of H_2_O_2_ into H_2_O. The 2,6-DMP radicals’ dimerization allows the hydrocoerulignone formation, then is oxidized by LPMO to the chromogenic coerulignone, that shows maximum absorbance at 469 nm [17,18].

The enzymatic assays using crude extracts of DC40, DC41, DC50 and DC51 transplastomic, wild-type (Nt) and PRV control plants, obtained with the acetone precipitation procedure, demonstrated LPMO activity. For all tested samples, except those corresponding to the control plants, an increase in activity (mU/mL), in a dose-dependent manner, was observed (Figure 7). In terms of specific activity, DC40 and DC50 reached the values of 0.11 ± 0.03 and 0.10 ± 0.01 U·g^−1^ of total soluble protein, respectively, indicating a low oxidase activity. Interestingly, the recombinant AA9 LPMO without the N-terminal modification showed higher specific activity values and in particular, DC41 and DC51 revealed 0.64 ± 0.02 U·g^−1^ and 0.26 ± 0.01 U·g^−1^, resulting 5.8 and 2.6-fold higher than those detected for their respective counterparts. Moreover, on the basis of the different expression levels for plastid-based AA9 LPMOs, evaluated by Western blot analysis, for DC40 and DC50 can be estimated specific activities of 16.5 ± 1.0 and 15.1 ± 1.5 U·g^−1^ AA9 LPMO. Interestingly, the recombinant proteins without the N-terminal fusion, DC41 and DC51, revealed values of 64.6 ± 2.7 U·g^−1^, and 51.2 ± 1.8 U·g^−1^, resulting 3.9 and 3.4-fold higher than those of their respective counterparts. Further, the specific activity of TaAA9B in DC41 was 1.3-fold higher than the specific activity of TrAA9B in DC51 protein extracts.

## 3. Discussion

Lignocellulosic biomass is the most abundant renewable resource in nature suitable for biofuel production. However, the poor efficiency of enzymatic hydrolysis of some recalcitrant lignocellulosic materials remains a key limiting step in their processing [19]. In recent years, great attention has been given to the auxiliary enzyme family 9 (AA9, formerly GH61) due to its booster activity in cellulose enzymatic degradation [20,21].

In the present study, we explored the feasibility of the production of two AA9 LPMO proteins by tobacco plastid transformation, since, previously, this approach enabled very encouraging results in terms of protein yield, hydrolytic activity, and stability of recombinant cellulolytic enzymes [14]. TrAA9B from *T. reesei* was chosen as this filamentous fungus is one of the best-studied organisms used in biomass conversion; in fact, it produces a multitude of (hemi)cellulases and is the best known and commercially used cellulolytic system [20]; TaAA9B from *T. aurantiacus*, instead, was selected for its putative thermophilic properties. Neither TrAA9B nor TaAA9B were previously expressed in planta. It is well known that many factors may affect protein yield and stability in plastids. Among them, the choice of regulatory elements at N-terminal of transgene may have a strong impact on accumulation of recombinant proteins [12,22]. Thus, we used two N-terminal regulatory sequences for both *TrAA9B* (i.e., DC50 and DC51) and *TaAA9B* (i.e., DC40 and DC41) genes: a 5′ untranslated region from *T7g10* in DC41 and DC51 plants, and a 5′ translation control region (5′-TCR), containing the 5′-UTR and the first 14 amino acids (DB) of the plastid *atpB* gene, in DC40 and DC50 plants. For *T. reesei* AA9 LPMO, the highest protein yield (5% TSP) was obtained in DC50 transplastomic plants, containing the AtpB-TrAA9B fusion protein, while the use of *T7g10* 5′-UTR in DC51 plants produced a protein accumulation level of 0.5% TSP. The highest expression detected in DC50 transplastomic plants was linked to a mild mutant phenotype mainly due to a growth retardation, since no alteration was observed in all other transplastomic plants expressing AA9 LPMO at lower level. This effect should be ascribed to intrinsic properties of TrAA9B, because when the same protein was accumulated at a lower level (DC51 plants), no mutant phenotype was observed. Phenotypic effects are often reflected in a reduction of the accumulation of Rubisco protein, visible in the SDS-PAGE profile, compared to controls. DC50 plant protein extracts, as well as those of the other transplastomic plants, did not show any alteration in their profiles compared to PRV control plants (Appendix A). Different results were obtained with TaAA9B protein, since the use of the *T7g10* 5′-UTR and *atpB* 5′-TCR regulatory sequences yielded similar accumulation levels in DC41 (1% TSP) and DC40 (0.7% TSP) transplastomic plants. Likewise, the use of three different DB regions (i.e., *tetC*, *nptII*, *gfp* DB) produced variable expression levels for an endoglucanase (*cel6A*) and a β-glucosidase (*bglC*) from *Thermobifida fusca* [23,24]. Particularly for Cel6A, the DB sequence originated from the *tetC* gene was the best choice, while *nptII* DB yielded the highest accumulation level for *bglC* expression. However, the variable expression level detected in our transplastomic plants expressing AA9 LPMO enzymes could not be due solely to the regulatory sequences used, but it may also be influenced by the intrinsic properties and/or the microbial origin of the recombinant proteins. For example, Kolotilin et al. [25] demonstrated that the best expression cassette for the bacterial xylanase XynA from *Clostridium cellulovorans* gave contrasting results for the accumulation of two fungal xylanases, Xyn10A and Xyn11B, from *Aspergillus niger* [25]. In a similar way, the 5′-TCR of *atpB* yielded 0.4% TSP for the endoglucanase from *Sulfolobus solfataricus*, and a ≥75% TSP accumulation of the β-glucosidase from *Pyrococcus furiosus* [14], the highest expression level achieved by chloroplast transformation so far. Altogether, these results demonstrate that empirical selection of regulatory sequences is required to optimize, for each gene of interest, the expression in transplastomic plants [14,24].

The use of 5′-TCR of the plastidial AtpB protein in DC50 transplastomic plants not only increased the overall protein accumulation but appeared to also protect the recombinant protein from degradation in older leaves of mature plants. Differently, the same regulatory sequence in DC40 plants had neither improved recombinant protein yield nor its in planta stability since no accumulation was detected in older leaves suggesting that TaAA9B was susceptible to proteolytic degradation, as already observed for rotavirus VP6 and HIV1-Pr55^gag^ proteins [26,27]. The use 5′-UTR of *T7g10* coding sequence in DC41 and DC51 transplastomic plants detected the recombinant proteins in mature leaves (fifth and seventh), suggesting that either translation or protein stability can limit foreign protein accumulation in the transplastomic lines as already found for some viral antigens [28,29].

Despite the absence of a specific assay to test the enzymatic activity of the LPMO family, a spectrophotometric method described by Breslmayr, et al. [17] was used to measure the peroxidase activity of plastid-based AA9 LPMO crude extracts. Since this enzymatic assay failed with the protein obtained with the standard extraction protocol due to the interference of colored pigments, it has been necessary to set up an alternative protein extraction method to remove these contaminations. The obtained uncolored crude protein samples showed the highest enzymatic activity in DC41 and DC51 transplastomic plants that reached 64.6 ± 2.7 and 51.2 ± 1.8 U·g^−1^ AA9 LPMO, respectively, whereas a 3.9- and 3.4-fold lower activity was estimated for DC40 and DC50 protein extracts (16.5 ± 1.0 and 15.1 ± 1.5 U·g^−1^ AA9 LPMO, respectively). Structural and characterization analyses revealed that AA9 auxiliary family are copper-dependent enzymes and that the Cu^2+^ ion is coordinated by a motif denoted as histidine brace (His-brace) formed by the highly conserved N-terminal histidine and a second histidine later in the sequence [30]. The His-brace motif is strictly conserved in all the LPMOs studied to date [31,32] and is considered essential for LPMO activity [8,33,34,35]. Since, in our DC40 and DC50 transplastomic plants, the AA9 LPMO proteins have the N-terminal “blocked” by the fusion with the first 14 amino acids from the plastid AtpB protein, the lower enzymatic activity observed, compared to DC41 and DC51 counterparts (without N-terminal fusion), may be due to this modification. Although the “naked” plastid-based AA9 (DC41 and DC51) were produced at low protein yield, the predicted specific activities resulted in being higher than those measured for other AA9 enzymes using the same spectrophotometric assay, even if a direct comparison is not feasible due to different sample purity (crude extracts vs. purified recombinant proteins). In particular, Breslmayr, et al. [17] found the *Neurospora crassa* NcLPMO9C and NcLMPO9F activities to be 32.2 and 2.2 U·g^−1^, respectively, while *Myriococcum thermophilum* was 30.9 U·g^−1^. Similarly, de Gouvêa et al. [36] and Guo et al. [37] reported a peroxidase activity of 8.33 U·g^−1^ for the AfAA9_B LPMO of *Aspergillus fumigatus*, and 17 U·g^−1^ for the LPMO from *Myceliophthora thermophila* C1 (MtC1LPMO), respectively.

Our results demonstrated the suitability of plastid transformation as a platform for heterologous production of AA9 LPMO proteins and indicate that to retain their oxidative activity, these auxiliary enzymes need to be expressed without any N-terminal fusion. Despite the stable production of plastid-based AA9 family proteins being demonstrated in all the plant generations tested, further investigation is necessary to increase the protein yield obtained in this work and to carry out an in-depth evaluation of their booster activity in biomass digestion. To this aim, better results could be achieved by using alternative 5′-UTRs and/or bacterial LPMOs coding sequences, since it has been largely proved that chloroplasts, due to their endosymbiotic origin, can successfully produce recombinant proteins from bacterial sources [14,16,38,39,40,41].

An improved production of plastid-based auxiliary enzymes, in terms of protein yields and specific activity, still represents an important chance to promote a sustainable and optimized biomass hydrolysis. Indeed, one major limitation of enzyme application is due to their high costs of production, that could be strongly reduced using chloroplast transformation technology as a green and renewable platform.

## 4. Materials and Methods

### 4.1. Plant Material and Growth Conditions

Plants of *Nicotiana tabacum* L. cv. Petite Havana for plastid transformation were grown in sterile conditions on hormone-free medium containing MS salts and B5 vitamins (Duchefa, Haarlem, The Netherlands), 30 g·L^−1^ sucrose and 8 g·L^−1^ agar, pH 5.6, at 24 °C with a 16 h photoperiod of 40 μmol photons m^−2^·s^−1^.

Homoplasmic transplastomic lines were rooted and propagated on medium containing MS salts with B5 vitamins, 30 g·L^−1^ sucrose, 0.1 mg·L^−1^ NAA, 80 g·L^−1^ agar and with 500 mg·L^−1^ of spectinomycin under controlled conditions, and subsequently transferred to soil in a growth chamber (14 h light, 200 μmol photons m^−2^·s^−1^, at 25 °C, and 10 h dark at 20 °C) for seeds production.

Seeds derived from transplastomic plants transformed with either *TaAA9B* or *TrAA9B* genes (DC series) or with the empty control vector (PRV) were sown in vitro under controlled conditions (16 h light 40 μmol photons m^−2^ s^−1^ and 8 h dark at 24 °C) on Murashige and Skoog (MS) medium with B5 vitamins (Duchefa), solidified with 0.8% (*w*/*v*) agar, with 20 g·L^−1^ sucrose and 500 mg·L^−1^ of spectinomycin, or they were sown in soil in a growth chamber (14 h light, 200 μmol photons m^−2^·s^−1^, at 25 °C, and 10 h dark at 20 °C).

### 4.2. Genes Design and Construction of Plastid Transformation Vectors

The *TrAA9B* and *TaAA9B* genes from *Trichoderma reesei* (GenBank accession number AY281372.1) and *Thermoascus auranticus* (GenBank accession number KF170230.1), respectively, were synthetized by GeneArt (ThermoFisher Scientific, Waltham, MA, USA).

The synthetic *TaAA9B* and *TrAA9B* genes were digested with *NheI* and *XbaI* restriction enzymes and cloned into pHK30 and pHK40 plasmids [42,43] replacing the *neo* coding gene to develop pDC40/41 and pDC50/51 vectors, respectively.

The expression cassettes pDC41 and pDC51 contain the *rrn* promoter, the 5′-UTR of *E. coli* phage T7 gene *10* and the plastid *rbcL* gene 3′-UTR, whereas in pDC40 and pDC50 constructs the *rrn* promoter was fused with 5′ translation control region (5′-TCR) that includes the 5′-UTR and 42 N-terminal nucleotides (DB) of the plastid *atpB* gene (Figure 1).

### 4.3. Stable Plastid Transformation and Southern Blot Analysis

DNA from pDC vectors were extracted by Plasmid Maxi Kit (Qiagen, Hilden, Germany) and used for delivery in tobacco leaf tissue. Biolistic experiments, in vitro regeneration and a selection of shoots were carried out according to the protocol described in Scotti and Cardi [44].

### 4.4. Expression Analysis

A semiquantitative RT-PCR analysis (27 cycles) was performed to verify the expression of recombinant proteins. Total RNA from the leaf of transplastomic and control was extracted with the RNeasy^®^ Plant Mini kit (Qiagen). cDNA was synthetized using the RevertAid RT Reverse Transcription kit (Thermo Scientific, Waltham, MA, USA) following the manufacturer’s instructions. Gene specific RT-PCR primers (Table 1) were designed using the online available tool Primer-BLAST.

### 4.5. Protein Extraction and Western Blot Analysis

Leaves from transplastomic and control plants were grinded in liquid nitrogen and total proteins were extracted by homogenization in buffer A (TrisCl 100 mM pH 7.8 containing 0.2 M NaCl, 1 mM EDTA, 2% (*v*/*v*) SDS, 0.2% (*v*/*v*) Triton X100, 2% (*v*/*v*) 2-mercaptoethanol, 1 mM PMSF, 1× protease inhibitor cocktail (Sigma, St. Louis, MO, USA) (leaves (g): buffer A (mL) ratio 1:3). The homogenate was centrifuged at 16,000× *g* and 4 °C for 20 min and supernatant recovered. Samples were electrophoresed either in a 10% or 12% SDS-PAGE and blotted onto Hybond ECL nitrocellulose membrane (GE Healthcare, Chicago, IL, USA). Recombinant FlagBAP protein (Sigma) was used as control. Membranes were incubated with a 1:4000 dilution of monoclonal anti-Flag M2 antisera (Sigma) and subsequently with an HRP-conjugated anti-mouse antibody (1:70,000, 1 h at RT). Chemiluminescent signal was measured using a ChemiDoc^TM^ XRS+ and images were analysed using the Image Lab^TM^ Software (Bio-Rad, Hercules, CA, USA).

Crude protein samples for enzymatic assays were extracted using acetonic precipitation. Briefly, leaves from transplastomic and control plants grinded in liquid nitrogen were homogenated in 80.8% acetone containing 0.2% 2-mercaptoethanol (solution B) (leaves (g):solution B (mL) ratio 1:10), incubated on ice for 15 min, stirred for 1 h at 4 °C and centrifuged at 20,000× *g* for 15 min. Following a wash in the same solution, pellet was resuspended in acetone containing 0.07% 2-mercaptoethanol, 2 mM EDTA, 1 mM PMSF (solution C), incubated on ice for 15 min, stirred for 1 h at 4 °C and centrifuged at 20,000× *g* for 15 min. Pellet was then washed with solution C and then resuspended in buffer A (pellet (g): buffer A (mL) ratio 1:10). Following incubation on ice for 15 min, stirring for 1 h at 4 °C and centrifugation, supernatant was recovered and dialyzed three times against 20 vol 20 mM Na phosphate buffer pH 7.5.

### 4.6. Spectrophotometric LPMO Activity Assays

The crude extracts of DC40, DC41, DC50, DC51 transplastomic plants, and PRV and wild-type (Nt) plants used as controls, obtained with the acetonic precipitation, were used for the enzymatic assay. The LPMO activity was measured using 2,6-dimethoxyphenol (2,6-DMP) and H_2_O_2_, as described by Breslmayr, et al. [17]. The reaction mixture containing Na phosphate buffer (100 mM, pH 7.5), 10 mM 2,6-DMP and 5 μM hydrogen peroxide was incubated for 15 min at 30 °C. After the addition of different amounts of DC40, DC41, DC50 and DC51 extracts (25-50-75-100 µg of recombinant AA9 LPMO estimated by Western blot analysis) to the reaction mixture, the increase in absorbance was continuously recorded at 469 nm over a reaction time of 300 s at 30 °C. One unit of enzyme activity is defined as the amount of enzyme required to release 1μM of coerulignone (ε_469_ = 53,200 M^−1^ cm^−1^) per minute under the described conditions.

## Figures and Tables

**Figure 1 ijms-24-00309-f001:**
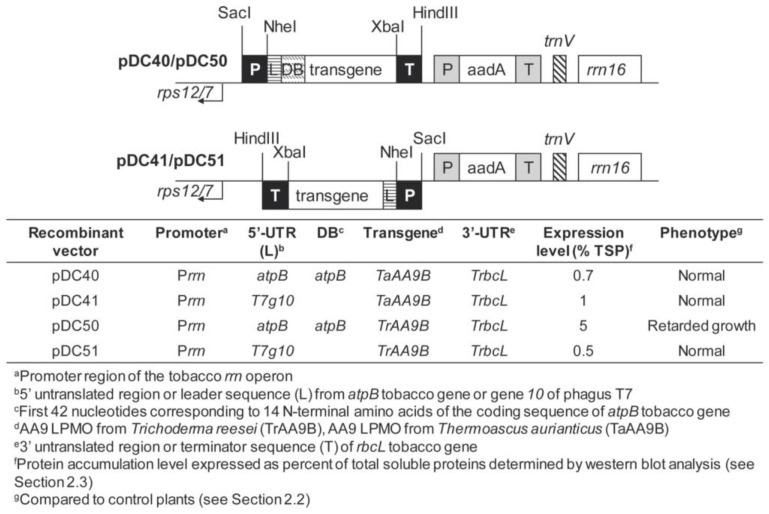
Schematic representation of plastid transformation vectors containing transgenes encoding the family AA9 of LPMOs from *Thermoascus aurantiacus* (TaAA9B) and *Trichoderma reesei* (TrAA9B). For each vector, regulatory sequences, protein accumulation level and phenotype in corresponding transplastomic plants are indicated.

**Figure 2 ijms-24-00309-f002:**
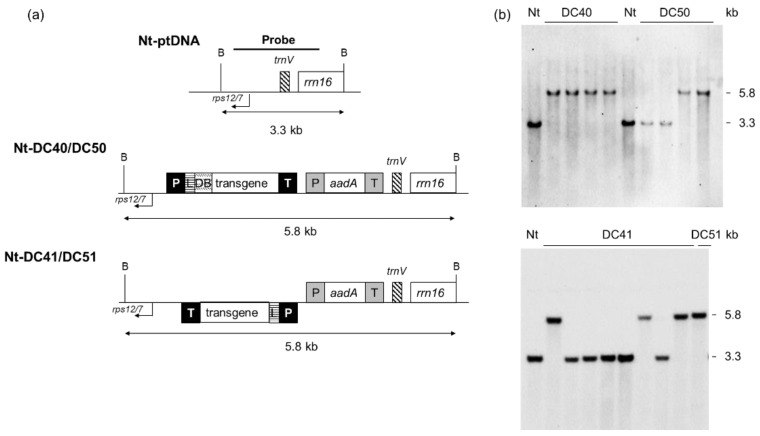
Identification of homoplasmic transplastomic tobacco lines for each construct. Schematic representation of the targeting region in the plastid genome (Nt-ptDNA) and maps of transformed (Nt-DC series) plastid genome regions involved in transgene integration. Below the maps are the indicated expected sizes of the *Bam*HI DNA fragments in Southern blot analyses. *rps12/7* and *trnV* correspond to the plastome integration site of the expression cassettes. (**a**) Selection of homoplasmic lines by Southern blot analyses. Examples of transplastomic lines obtained with each vector are reported. (**b**) TrAA9B, AA9 LPMO from *Trichoderma reesei*; TaAA9B, AA9 LPMO from *Thermoascus aurantiacus*; *P*, promoter; *L*, 5′-UTR or leader sequence; DB, downstream box corresponding to the first 42 N-terminal nucleotides of native *atpB* tobacco coding sequence; *aadA*, *aminoglycoside 3*′ *adenylyltransferase* selectable marker gene; T, terminator sequence; B, BamHI restriction site.

**Figure 3 ijms-24-00309-f003:**
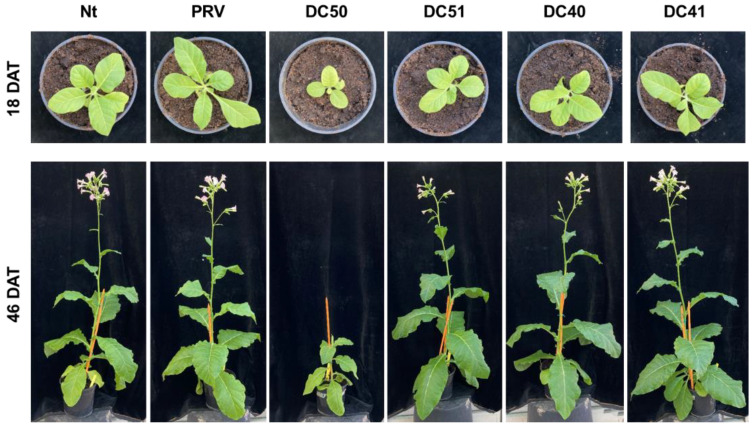
Plant phenotype. Greenhouse-grown transplastomic plants expressing TrAA9B (DC50 and DC51) and TaAA9B (DC40 and DC41) proteins compared to controls (Nt and PRV, wild type and transformed with the empty vector, respectively) at 18 and 46 days after transplanting (DAT).

**Figure 4 ijms-24-00309-f004:**
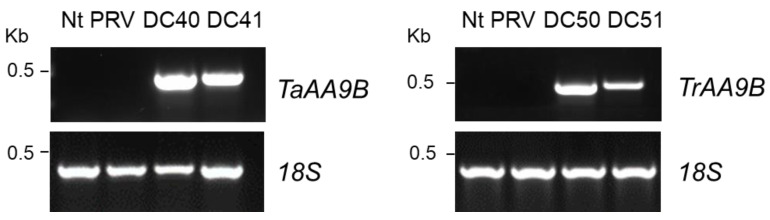
Representative examples of semi-quantitative RT-PCR (27 cycles) analysis of selected T2 transplastomic plants to verify the expression of TaAA9B and TrAA9B transgenes. DC40 and DC41 transplastomic plants expressing AA9 LPMO from Thermoascus aurantiacus (TaAA9B); DC50 and DC51 transplastomic plants expressing AA9 LPMO from Trichoderma reesei (TrAA9B); Nt and PRV control plants, wild type and transformed with the empty vector, respectively.

**Figure 5 ijms-24-00309-f005:**
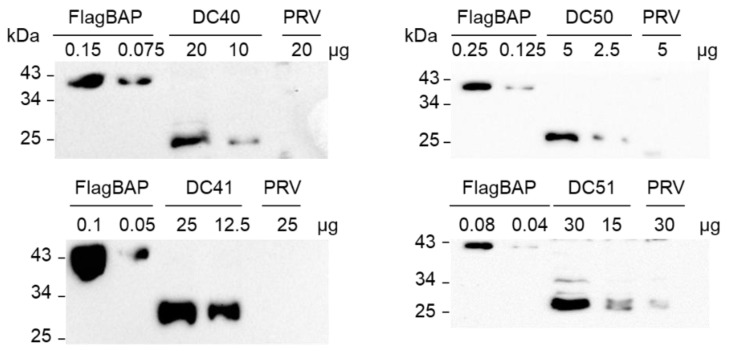
Detection of TaAA9B (DC40 and DC41 transplastomic plants) and TrAA9B (DC50 and DC51 transplastomic plants) recombinant proteins by Western blot analysis. Each transplastomic line shows a protein band corresponding in size to the recombinant protein. PRV, control plants transformed with an empty vector; FlagBAP, N-terminal FLAG fusion protein of *E. coli* bacterial alkaline phosphatase.

**Figure 6 ijms-24-00309-f006:**
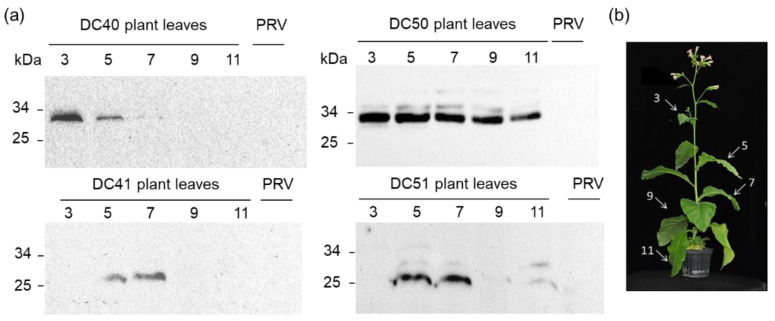
In planta stability of AA9 LPMO recombinant proteins accumulated in DC40/DC41 (TaAA9B) and DC50/DC51 (TrAA9B) transplastomic plants. Protein accumulation level in leaves at different development stages by Western blot analysis (**a**). Arrows indicate leaves sampled for protein analyses, where 3 indicates the youngest leaf, 11 the oldest one (**b**).

**Figure 7 ijms-24-00309-f007:**
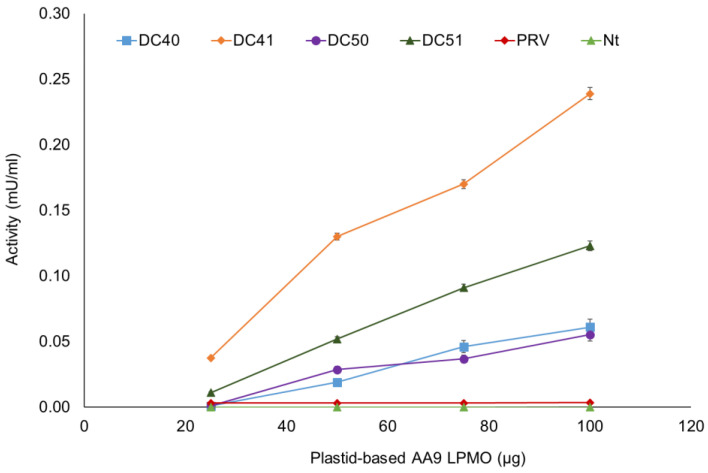
Profile of LPMO activity (mU/mL) of crude protein extracts from transplastomic and control plants. Assayed amounts of plastid-based AA9 LPMOs (25–100 µg) were estimated by Western blot analysis. All data are expressed as mean values (±SD), from three independent biological/technical replicates.

**Table 1 ijms-24-00309-t001:** List of primers used for semi-quantitative RT-PCR analysis.

	Primer Sequence (5′-3′)
DC40/DC41	For CATCGCCTGGTCTACTACGG
	Rev ATCGTGGTAGAGTGCCGTTC
DC50/DC51	For GCTTCATCTCCCCTGACCAA
	Rev AGGGATGGTGTAGCTCGTGA
18S	For TAGATAAAAGGTCGACGCGG
	Rev CCCAAAGTCCAACTACGAGC

## Data Availability

Not applicable.

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
