# Peer review of "Tobacco Plastid Transformation as Production Platform of Lytic Polysaccharide MonoOxygenase Auxiliary Enzymes"

_ijms, 2022, doi:10.3390/ijms24010309_

Round 1
Reviewer 1 Report
The present manuscript of Tamburino et al. addresses a quite relevant topic in the plant biotechnology field: the production of lytic polysaccharide monooxygenases (LPMOs) in transplastomic tobacco, which is an important component in biomass conversion. For expression, they chose microorganismic enzymes TrAA9B and TaAA9B.
For optimization of expression they tested different N-terminal 5´UTRs and Downstream boxes. Since expression levels of transplastomic proteins are decisive for the future use of chloroplast transformation technique, this methodology has high practical relevance.
The article is written in a clear and concise style.
Concerning the aspect of novelty, the group already has published a strong paper on cellulolytic enzymes in tobacco (Castiglia et al. 2016).
From the presentation of the data, it becomes obvious that the experiments have been carried out with special care. They cover the complete procedure of protein expression through plastid transformation, genetic analysis, Western analysis of transformants on protein expression and its characterization by spectrophotometric activity assay.
In conclusion, the paper presents a good deal of novel findings and showcases a useful approach that enables scientific progress on both fields: plastid transformation and enzymatic biomass conversion. I recommend it for publication without fault-finding.
Lit.:
Castiglia et al. Biotechnol Biofuels (2016) 9:154, DOI 10.1186/s13068-016-0569-z
Author Response
Reviewer 1
The present manuscript of Tamburino et al. addresses a quite relevant topic in the plant biotechnology field: the production of lytic polysaccharide monooxygenases (LPMOs) in transplastomic tobacco, which is an important component in biomass conversion. For expression, they chose microorganismic enzymes TrAA9B and TaAA9B.
For optimization of expression they tested different N-terminal 5´UTRs and Downstream boxes. Since expression levels of transplastomic proteins are decisive for the future use of chloroplast transformation technique, this methodology has high practical relevance.
The article is written in a clear and concise style.
Concerning the aspect of novelty, the group already has published a strong paper on cellulolytic enzymes in tobacco (Castiglia et al. 2016).
From the presentation of the data, it becomes obvious that the experiments have been carried out with special care. They cover the complete procedure of protein expression through plastid transformation, genetic analysis, Western analysis of transformants on protein expression and its characterization by spectrophotometric activity assay.
In conclusion, the paper presents a good deal of novel findings and showcases a useful approach that enables scientific progress on both fields: plastid transformation and enzymatic biomass conversion. I recommend it for publication without fault-finding.
Lit.:
Castiglia et al. Biotechnol Biofuels (2016) 9:154, DOI 10.1186/s13068-016-0569-z
R: We thank the reviewer for positive comments.
Reviewer 2 Report
Dear authors,
i have gone through the manuscript and find the work is fine. I have some suggestions to improve the work and representation. Please find the suggestions below:
1. First and farmost, it is really hard to find the significance of work as higher expression of cellulolytic enzymes degrade the cellulose and increase the available sugars but it also reduce the mechanical strength of cells along with making them prone to degradation. Then what is the purpose and future prospect of work?
2. Methodology is well defined,
3. In section 2.2 it is mentioned that DC50 have retarded growth pattern. why retarded growth? what is the effect of gene transfer on other genes? How much time DC50 took to reach maturity?
4. Section 2.4, Add reference for mechanism of assay.
Specific activity was higher in DC41 was higher than DC51. It is because of lower expression of other proteins of higher enzyme activity. If it is due to lower expression of other proteins, which kind of proteins were expressed poorly?
Once again, add significance of work in abstract and main body.
Good luck.
Author Response
Reviewer 2
Dear authors,
I have gone through the manuscript and find the work is fine. I have some suggestions to improve the work and representation. Please find the suggestions below:
- First and farmost, it is really hard to find the significance of work as higher expression of cellulolytic enzymes degrade the cellulose and increase the available sugars but it also reduce the mechanical strength of cells along with making them prone to degradation. Then what is the purpose and future prospect of work?
R: we thank the reviewer for the comment, but we think there is a misunderstanding because the aim of our work is the use of chloroplast as biofactory to produce recombinant LPMOs. Recombinant enzymes produced by plastid transformation are confined into the organelle and no export from chloroplasts has never been demonstrated. Although the in situ degradation of cellulose and mechanical strength of cells have never been measured in transplastomic plants expressing several cellulolytic enzymes at higher level compared to LMPOs of this work, we assume that these enzymes cannot affect such properties due to their containment into the chloroplast. However, to clear this issue we changed accordingly both the abstract and the introduction by emphasizing the relevance to develop a low-cost and sustainable platform of production of LPMO enzymes in order to concretize biomass bioconversion into fermentable sugars.
Lines 18-21 “Here, we report for the first time, the production of the AA9 LPMOs from the mesophilic Trichoderma reesei (TrAA9B) and the thermophilic Thermoascus aurantiacus (TaAA9B) microorganisms in tobacco by plastid transformation with the aim to test this technology as cheap and sustainable manufacture platform.”
Lines 75-78 “The aim of this study was to explore, for first time, the feasibility of plants and, in particular, chloroplast transformation technology as cheap and sustainable platform to produce in tobacco two synthetic Lytic Polysaccharide MonoOxygenases, belonging to AA9 family and derived from the mesophilic Trichoderma reseei (termed herein TrAA9B) and the thermophilic Thermoascus auranticus (termed herein TaAA9B) fungi. Indeed, the availability of a green, low-cost and sustainable manufacture platform will allow to overcome one of the major limitations in application of enzymes in biomass conversion reducing production and processing costs.”
- Methodology is well defined,
R: we thank the reviewer for the comment.
- In section 2.2 it is mentioned that DC50 have retarded growth pattern. why retarded growth? what is the effect of gene transfer on other genes? How much time DC50 took to reach maturity?
R: The retarded growth observed in DC50 plants is due to the high expression of TrAA9B, because DC51 plants, expressing the same protein at lower level, did not show any mutant phenotype. Phenotypic effects are often reflected in a reduction in the accumulation of Rubisco protein compared to control plants that can be observed in the SDS-PAGE profile. Protein profile of DC50 plants, as well as those of the other transplastomic plants, did not show any alteration compared to PRV control plants, as it can be observed in the Supplementary Figure S1 (it was added in the revised text).
Further, no position effect or gene silencing have been observed in chloroplast transformation method, because the transgene is inserted in a precise region of the plastid genome, being mediated by two homologous recombination events.
DC50 plants reach to maturity 90 days after transplanting, we inserted this information at line 126.
To clear this issue, we added the following sentences in lines 229-234: “This effect should be ascribed to intrinsic properties of TrAA9B, because when the same protein was accumulated at lower level (DC51 plants) no mutant phenotype was observed. Phenotypic effects are often reflected in a reduction of the accumulation of Rubisco protein, visible in the SDS-PAGE profile, compared to controls. DC50 plant protein extracts, as well as those of the other transplastomic plants, did not show any alteration in their profiles compared to PRV control plants (Supplementary Figure S1).”
- Section 2.4, Add reference for mechanism of assay.
R: In agreement with the reviewer, the cited reference (Breslmayr, E.; Hanžek, M.; Hanrahan, A.; Leitner, C.; Kittl, R.; Šantek, B.; Oostenbrink, C.; Ludwig, R. A fast and sensitive activity assay for lytic polysaccharide monooxygenase. Biotechnol Biofuels 2018, 11, 79, doi:10.1186/s13068-018-1063-6) gives a complete description of the mechanism of the assay that utilize 2,6 DMP and H2O2 as substrate and co-substrate, respectively. As the same mechanism is also reported in Breslmayr, E., Daly, S., Požgajčić, A. et al. Improved spectrophotometric assay for lytic polysaccharide monooxygenase. Biotechnol Biofuels 2019,12, 283, doi.org/10.1186/s13068-019-1624-3, we added both references in line 180.
- Specific activity was higher in DC41 was higher than DC51. It is because of lower expression of other proteins of higher enzyme activity. If it is due to lower expression of other proteins, which kind of proteins were expressed poorly?
R: Based on Coomassie blue staining of polyacrylamide gel of LPMOs accumulated in transplastomic DC plants (Supplementary Figure S1) we did not find any differences between transplastomic plants expressing recombinant LPMOs and control plants. Furthermore, in the enzymatic assay, the same amounts of recombinant AA9 LPMOs, estimated by western blot analysis, were used for both DC41 and DC51. Thus, we can suggest that higher specific activity measured with crude extracts from DC41plants than those observed with extracts from DC51 plants is likely due to higher activity of recombinant TaAA9B protein (in DC41 plants) compared to TrAA9B protein (in DC51 plants).
Once again, add significance of work in abstract and main body. R: done.
Good luck.
Round 2
Reviewer 2 Report
Dear author,
The comments have been addressed very well. However I have just one suggestion, as you mention the overexpressed enzyme remain confined in organelle. Then what is the use of over expression.
Instead, design something that secrete enzyme outside so that the plat cells can be used as real reactor and enzyme can be recovered with sacrificing the plants.
Good luck